# Real-World Effectiveness in Hypertension and Hyperlipidemia Collaborative Management between Pharmacies and Primary Care in Portugal: A Multicenter Pragmatic Controlled Trial (USFarmácia^®^)

**DOI:** 10.3390/ijerph20156496

**Published:** 2023-08-01

**Authors:** Suzete Costa, José Luís Biscaia, Maria Rute Horta, Sónia Romano, José Guerreiro, Peter Heudtlass, Maria Cary, Mariana Romão, António Teixeira Rodrigues, Ana Miranda, Ana Paula Martins, Ana Sofia Bento, João Pereira, Céu Mateus, Dennis K. Helling

**Affiliations:** 1NOVA National School of Public Health (ENSP), Universidade NOVA de Lisboa, 1600-560 Lisboa, Portugal; 2Institute for Evidence-Based Health (ISBE), 1649-028 Lisboa, Portugal; 3Faculdade de Medicina, Universidade de Lisboa, 1649-028 Lisboa, Portugal; 4USF São Julião da Figueira, Agrupamento dos Centros de Saúde (ACeS) do Baixo Mondego, 3080-134 Figueira da Foz, Portugal; 5Centre for Medicines Information and Health Interventions (CEDIME), Infosaúde, Associação Nacional das Farmácias, 1249-069 Lisboa, Portugal; 6Centre for Health Evaluation & Research (CEFAR), Infosaúde, Associação Nacional das Farmácias, 1249-069 Lisboa, Portugal; 7Registo Oncológico Nacional, Instituto Português de Oncologia de Lisboa Francisco Gentil, 1099-023 Lisboa, Portugal; 8Pharmacy, Pharmacology & Health Technologies, Faculdade de Farmácia, Universidade de Lisboa, 1649-003 Lisboa, Portugal; 9Public Health Research Centre (PHRC/CISP), Comprehensive Health Research Centre (CHRC), 1600-560 Lisboa, Portugal; 10Health Economics at Lancaster, Division of Health Research, Lancaster University, Lancaster LA1 4YX, UK; 11Skaggs School of Pharmacy and Pharmaceutical Sciences, University of Colorado, Denver, CO 80045, USA

**Keywords:** community pharmacy, primary care, pharmacy services, collaborative, hypertension, hyperlipidemia, trial, effectiveness, real-world, public health interventions

## Abstract

There is evidence of the efficacy of collaborative health interventions with pharmacies and primary care providers but little of its real-world effectiveness. We aimed to assess the effectiveness and discuss the design and challenges of hypertension and hyperlipidemia management between pharmacies and primary care providers using real-world data exchange between providers and experimental bundled payment. This was a pragmatic, quasi-experimental controlled trial. We collected patient-level data from primary care prescription claims and Electronic Medical Record databases, a pharmacy claims database, and patient telephone surveys at several time points. The primary outcomes were changes in blood pressure and total cholesterol. We used matched controls with difference-in-differences estimators in a Generalized Linear Model (GLM) and controlled interrupted time series (CITS). We collected additional data for economic and qualitative studies. A total of 6 Primary Care Units, 20 pharmacies, and 203 patients entered the study. We were not able to observe significant differences in the effect of intervention vs. control. We experienced challenges that required creative strategies. This real-world trial was not able to show effectiveness, likely due to limitations in the primary care technology which affected the sample size. It offers, however, valuable lessons on methods, strategies, and data sources, paving the way for more real-world effectiveness trials to advance value-based healthcare.

## 1. Introduction

The prevalence of hypertension in the Portuguese population in 2013 was 29.1% and 64.4% did not have adequate treatment and control [1]. The prevalence of high blood cholesterol was 52% (50.0–54.8%) in 2015 [2]. The PRECISE study, from 2019, reports that 43.3% of hypertensive patients are not controlled for blood pressure, 82.1% of hypertensive patients present high blood cholesterol, and 62.3% of those are not controlled for blood cholesterol [3]. Hypertension and hyperlipidemia, if poorly managed, can lead to cardiovascular disease, including heart failure and hypertensive heart disease.

Most of these patients are usually followed in primary care by general practitioners (GPs) in Portugal. Cardiovascular disease and cancer are the leading causes of death in Portugal. The country lags behind Spain, Italy, and France in preventable mortality, suggesting that more could be done to save lives by reducing risk factors for cancer and cardiovascular disease [4].

In 2013, 12.3% (15.7% in 2017) of all hospital admissions in Portugal were due to Ambulatory Care Sensitive Conditions (ACSCs) and 93.7% followed an emergency room visit. In 2013, the third and fourth most common ACSCs were heart failure and hypertensive heart disease (second and sixth in 2017) [5,6]. Both studies show the immense opportunity to improve local service delivery.

The need to bring healthcare closer to primary, community, and self-care requires health governance aligned with evidence-based decisions [7,8] using integrated care following the Kaiser pyramid care model. In this model, the population is stratified according to the complexity of their condition using the most effective and least expensive resources and multi-professional teams [9].

There is evidence of improvements in health outcomes associated with certain public health services provided by pharmacists within an appropriate collaborative environment with physicians [10,11,12], including with cardiovascular risk, hypertension, and hyperlipidemia [10,13,14,15,16,17,18,19,20,21]. A meta-analysis of 39 randomized controlled trials comprising 14,224 patients demonstrated improvements in blood pressure (BP) following pharmacists’ interventions [20]. A quasi-experimental controlled trial in Portuguese pharmacies with 109 patients also achieved improvements in BP [22]. A systematic review of 21 randomized controlled trials which included a subset of 10 studies using 1196 patients demonstrated pharmacist-led improvements in total cholesterol (TC) [15]. Most studies were conducted in the USA, UK, and Canada with explicit interprofessional collaboration practices between pharmacists and physicians, often in integrated settings, such as managed care organizations and ambulatory clinics, and not in community pharmacies.

The vast majority of studies on pharmacy interventions are efficacy trials. They are designed to establish efficacy, e.g., if the intervention can work in well-resourced “ideal” settings, with highly selected and adherent populations, and using strict protocols. These trials are often paper-based, not integrated into the pharmacy’s daily workflow or software, not reimbursed, and limited to the outcomes of interest to researchers, all of which are relevant to establish proof of efficacy but not of effectiveness in real-world conditions under which we would expect these interventions to operate. Therefore, we have a vast body of literature seeking to establish efficacy for some already well-established efficacious pharmacy interventions.

By contrast, there are very few pragmatic controlled trials designed to establish effectiveness, e.g., does the intervention work when used in normal routine practice and with real-world patients, does it apply flexible protocols as in normal practice, is it integrated into the pharmacy’s daily workflow and software, is it reimbursed, and does it collect outcomes relevant to patients, providers, pharmacy owners, healthcare organizations, and payers? These studies provide more useful insights to help shape future successful pharmacy interventions in real-world conditions. We found two such research studies that seek to address the effectiveness of hypertension management in groups of community pharmacies [19,23].

Strategies to explain how pharmacy-based interventions may work are consistent with several behavioral theoretical models [24,25,26]. Wagner’s Chronic Care Model assisted by the Information-Motivation-Behavioral (IMB) framework used by Blalock was useful in guiding this collaborative intervention [10,27]. The Chronic Care Model predicts that improvements in six inter-related components—community resources, healthcare organization, self-management support, delivery system design, decision support, and clinical information systems—can produce changes in which informed, activated patients interact with prepared, proactive practice teams within primary care [27]. The theoretical framework used by Blalock is a modified version of the Information-Motivation-Behavioral Model, according to which the information and intervention package is expected to help providers and patients engage in care, subsequently leading to the adoption of behaviors that would translate into improved health outcomes [10].

This work is part of a wider research project on the economic evaluation of a pharmacy-based collaborative public health intervention with primary care providers [28,29,30,31]. Planning and conducting a well-designed trial for effectiveness and other dimensions should precede the economic evaluation of pharmacy-based public health interventions [29].

This trial aimed to assess the effectiveness and discuss the design and challenges of the first collaborative hypertension and hyperlipidemia management using real-world data exchange between pharmacies and a National Health Service (NHS) Primary Care Unit (PCU) and an experimental bundled payment in Portugal.

## 2. Materials and Methods

This trial was retrospectively registered with Current Controlled Trials (ISRCTN): ISRCTN13410498, on 12 December 2018: https://www.isrctn.com/ISRCTN13410498.

We developed a study protocol and statistical analysis plan and a Privacy Impact Assessment for this trial to conform with the General Data Protection Regulation (GDPR).

We used the Transparent Reporting of Evaluations with Nonrandomized Designs (TREND) of behavioral and public health interventions guidelines for study reporting [32] and the Template for Intervention Description and Replication (TIDieR) Checklist for describing the intervention [33].

### 2.1. Objectives

The primary objective was to test the hypothesis that collaborative management between pharmacies and primary care providers would result in improved outcomes (blood pressure and total cholesterol) over the usual care after the trial’s onset. A secondary objective was to explore the operation and discuss the lessons learned with this first collaborative trial.

### 2.2. Trial Design

This was a 6-month multicenter quasi-experimental controlled trial featuring domains of pragmatic trials [34,35,36,37].

The collaborative care model included Integrated Care Pathways (ICPs) and interprofessional meetings (Quality Circles). ICPs consisted of consensus-based clinical decision algorithms integrated into the pharmacy software. Quality Circles were based on principles of working relationships between pharmacists and physicians [38] and inspired by innovative experiences in the USA [39], Switzerland [40], and Australia [41].

The comparator (control group) was the usual care provided by other pharmacies and Primary Care Units.

### 2.3. Participants

#### 2.3.1. Eligibility Criteria for Intervention Primary Care Units and Pharmacies

Intervention PCUs and pharmacies were a coalition of willing providers aiming at working as an Integrated Patient Unit (IPU).

The intervention PCU expressed interest in this collaborative intervention, and we selected seven intervention pharmacies near the PCU, using the same dispensing software and data network provider, with a minimum pre-defined proportion of dispensing prescriptions from the intervention PCU active in the national Pharmacy Customer Loyalty Program based on the number of patients expected to be recruited per pharmacist while looking for a minimum feasible number of sites to facilitate trial management. These data were provided by the National Association of Pharmacies (ANF).

We adopted the NHS Patient Electronic Health Record (EHR) ‘Registo de Saúde Electrónico’ (RSE^®^) 2018 (SPMS EPE, Lisboa, Portugal) as the interface for data exchange between pharmacy dispensing claims software Sifarma^®^ 2000 2.9.4.3 (Glintt, Sintra, Portugal) and PCU Electronic Medical Record (EMR) software SClínico^®^ 2018 (SPMS EPE, Lisboa, Portugal). Before this trial, the Consent Form of the EHR RSE^®^ 2018 (SPMS EPE, Lisboa, Portugal) had no option for patients to consent to data access and exchange between pharmacists and other healthcare providers. The research team together with the NHS Shared Services of the Ministry of Health (SPMS) team developed two sentences for the patient to consent to the exchange of relevant health data (from pharmacists to GPs/nurses and from GPs/nurses to pharmacists) and added them to the Consent Form of the EHR RSE^®^ 2018 (SPMS EPE, Lisboa, Portugal) for the first time in Portugal.

Details and a snapshot of the added sentences can be seen in Appendix A.

#### 2.3.2. Eligibility Criteria for Control Primary Care Units and Pharmacies

We used prospective location-based controls to minimize the risk of contamination between intervention and controls and matched controls to minimize imbalance and improve causal inference. Matching was performed using cluster analysis at the (1) municipality level and at the (2) PCU level, as further explained.

Appendix A depicts the graphic representation of the cluster analysis used to estimate three-dimensional normalized Euclidean distance, using the statistical software R (2018) (The R Foundation for Statistical Computing, Vienna, Austria) [42], and used to rank potential best-match control municipalities on economic deprivation using the Municipality Purchasing Power Index (Índice de Poder de Compra Concelhio, IPCC): illiteracy, aging, and predominantly urban area. We used the R graphic plot [43].

Next, we mapped potential control PCUs in the top 30 municipalities and ranked them according to the best match criteria on PCU contractual type, PCU Electronic Medical Record software, PCU burden proxy, PCU patients/GP ratio, patients’ age, International Classification of Primary Care (ICPC-2) codes used by primary care GPs for diagnosis of hypertension and/or lipid metabolism disorder, and obesity. We then selected the best-match control PCUs.

Finally, we selected control pharmacies similar to intervention pharmacies.

We planned to invite 46 control pharmacies, aiming at obtaining at least 21 (46%), based on the average number of patients expected to be recruited per control pharmacy, as we expected recruitment to be more difficult.

See details in Appendix A.

#### 2.3.3. Patient Inclusion Criteria

We included adult patients of selected PCUs on medication for hypertension and/or hyperlipidemia (last prescription date within 6 months before baseline) that were also holders of an NHS number.

Patients could be either new to therapy (last prescription date within 2 months before baseline and no prescription date within 2–6 months before baseline) or usual medication users (last prescription date within 6 months before baseline and not eligible for new therapy) at baseline.

We planned to select uncontrolled patients at baseline (patients with blood pressure ≥140/90 mm Hg and/or total cholesterol ≥190 mg/dL). Pharmacists were instructed to measure baseline blood pressure and total cholesterol if no such measurement in the previous 30 days was available from primary care providers to enable recruitment and intervention.

Intervention patients had additional criteria related to the intervention package:Mobile phone users (to receive Short Message Service (SMS) refill reminders);Registered at the EHR RSE^®^ 2018 (SPMS EPE, Lisboa, Portugal) (to consent to data exchange between pharmacists and GPs);Members of the Pharmacy Customer Loyalty Program Saúda^®^ (to enable reimbursement to intervention pharmacies);Holders of a Patient Record in the pharmacy dispensing claims software Sifarma^®^ 2000 2.9.4.3 (Glintt, Sintra, Portugal) (to enable longitudinal tracking).

We excluded diabetic patients (although ICPs were in place for these patients in intervention pharmacies and PCUs for ethical reasons).

#### 2.3.4. Method of Recruitment

Eligible patients were recruited in pharmacies through case finding and prompt pharmacy software reminders upon presenting a prescription for antihypertensive or lipid-lowering medications.

Intervention pharmacists invited eligible patients with the aid of the Patient Information Leaflet (Appendix A). Diabetic patients were not excluded from the intervention due to ethical reasons but were excluded from this study as previously explained.

Due to the nature of this trial, patient written consent for the study was obtained following allocation [44]. We specified the nature of the consent [45] for intervention and data collection (intervention) and data collection (control) [46].

Pharmacists recorded the patient’s NHS number, name, mobile number, preferred time for the telephone survey, the reason for recruitment, and patient PCU in the pharmacy software. We captured these data daily to enable baseline telephone interviews within 24–48 h.

The pharmacy software generated an email to the intervention PCU to inform GPs upon each new patient enrolled.

#### 2.3.5. Settings and Locations

A list of intervention study sites is available. We do not disclose control sites for ethical reasons, as they were blinded, but we provide a list of their broad geographical locations (Appendix A).

### 2.4. Interventions

#### 2.4.1. Intervention

We developed ICPs (clinical decision algorithms for intervention pharmacists) and negotiated them with primary care providers before the trial. The technology team further embedded ICPs in the pharmacy software assisted by researchers. Intervention pharmacists followed the decision algorithms based on each patient’s situation at each visit. Referral criteria for primary care and timing of referral were explicitly defined in those ICPs. Partial delegation of tasks from primary care GPs to pharmacies was expected in the collaborative model. Since data exchange between settings was planned by design, feedback from GPs to the pharmacist would also feed the ICP to determine further actions at the pharmacy or direct requests sent to the GP for a non-scheduled medical appointment.

Hence, the collaborative intervention between pharmacies and primary care GPs was designed as patient-tailored according to ICPs in the pharmacy dispensing claims software Sifarma^®^ 2000 2.9.4.3 (Glintt, Sintra, Portugal) and data exchange with the primary care EMR software SClínico^®^ 2018 (SPMS EPE, Lisboa, Portugal) through the interface Patient EHR RSE^®^ 2018 (SPMS EPE, Lisboa, Portugal).

The designed collaborative complex and multifaceted health intervention package included the following innovative components: (1) point-of-care measurements; (2) cardiovascular risk assessment; (3) medication management; (4) request for repeat prescription; (5) lifestyle counseling; (6) referral and direct request from the pharmacy for a medical appointment, as per the ICP; (7) feedback from PCU and follow-up at the pharmacy, as per the ICP; (8) refill text reminder to the patient smartphone 10 days before the end of the last tablet in the last package; (9) Quality Circles; (10) report of a potential adverse drug event (ADE) (see Figure 1).

Components 1, 2, 3, and 5 are all part of hypertension and hyperlipidemia effective patient management as described in the literature. The collaborative intervention included an innovative feature: clinical decision algorithms for pharmacists (ICPs) to record results of components 1, 2, 3, and 5, referral criteria to enable a decision to refer to the GP or to continue monitoring, and data exchange between pharmacists and GPs’ software systems. Hence, ICPs require components 6 and 7. Data exchange between providers enabled component 4. Data communication to patients’ smartphones enabled component 8. Both components 4 and 8 enhance medication adherence, which is relevant to effectiveness. Component 9 was deemed relevant to enhance interprofessional engagement within the desired collaborative care model. Component 10 was added as a mandatory procedure in real-world effectiveness studies containing medicines.

A detailed description of the intervention component package can be seen in Appendix A.

Each pharmacy received a Project Promotional Pack and a Study Pack.

Pharmacists attended one training session and workshops accredited by the Portuguese Pharmaceutical Society. Pharmacists, researchers, and the project physician leader delivered the training.

Pharmacists, GPs, and nurses took part in regular Quality Circles involving a mix of team-building and joint training in the technology-based collaborative intervention package. All providers performed point-of-care measurements, cardiovascular risk assessment, and lifestyle counseling.

The frequency of pharmacy appointments was patient-tailored at pre-specified time intervals, as per the ICP, with some degree of flexibility.

The frequency of medical and nurse appointments was left to the discretion of providers to facilitate their endorsement yet hoping for some delegation to pharmacists.

We monitored the recruitment and intervention package through regular phone calls, on-site visits, and workshops with pharmacies and provided each pharmacy and PCU with a weekly Feedback Report (USFarmáciaWatch^®^) containing key performance indicators (Appendix A).

We used the Pharmacy Study Leaflet, training, trial monitoring, feedback to pharmacies, and reminders in software to call patients before scheduled visits to improve fidelity and promote patient retention and follow-up, emulating methods of clinical trials.

In the absence of NHS reimbursement of these services, the Pharmacy Customer Loyalty Program Saúda^®^ reimbursed intervention pharmacies. We designed an innovative billing and reimbursement model, simulating for the first time in Portugal an experimental risk–share bundled payment to mimic a hypothetical reimbursement scenario by the NHS inspired by the US Medication Therapy Management (MTM) Current Procedural Terminology (CPT) codes used by pharmacists for billing medication therapy management services [47] (Appendix A).

The intervention PCU had some incentives in place against quality targets, albeit non-financial, which included hypertension management (PCU model “A”).

#### 2.4.2. Usual Care

Control pharmacies recruited similar patients. We supplied a similar Study Pack, but no training was provided to preserve usual care, mirroring real-life practice and minimizing the risk of contaminating the comparator.

#### 2.4.3. Time Span

The study started with the recruitment of intervention pharmacies in March 2016 and concluded with database closure in November 2019. Patients were recruited in pharmacies between 1 May (first patient recruited) and 30 November 2018 (last patient recruited) and followed for at least 6 ± 2 months. The intervention was operational between May 2018 and the last follow-up in July 2019 (6 ± 2 months after the last recruited patient).

We allowed a run-in period (first 5 hypertension and/or hyperlipidemia patients per intervention pharmacy) to enable a learning curve [48] and these patients were included in the analysis.

Figure 2 provides a study flow diagram.

#### 2.4.4. Post-Trial Procedures and Ethical Considerations

Trial closure determined the closure of ICPs in the pharmacy software. Intervention pharmacies remained, however, equipped with paper ICPs provided in training to apply in their daily practice.

### 2.5. Outcomes

#### 2.5.1. Primary Outcomes

We used the changes in BP, TC, and the proportion of controlled patients 6 months after the onset of the trial. We defined outcome control as BP < 140/90 mmHg and TC < 190 mg/dL.

All intervention pharmacies used the same equipment and supplies to enhance measurement quality and ensure procedure standardization. As control pharmacies may or may not perform point-of-care measurements in their usual care, we did not provide the equipment and supplies, to avoid contaminating the usual care.

However, this study used blood pressure measurements and total cholesterol measured by primary care providers and extracted from the primary care Electronic Medical Records, as these were recorded in both intervention and control PCUs and available to researchers. Hence, these are outcomes as measured in real-world practice by GPs and nurses.

#### 2.5.2. Data Collection

We collected longitudinal patient-level data from the pharmacy claims database Sifarma^®^ 2000 2.9.4.3 (Glintt, Sintra, Portugal); primary care databases prescription claims PEM^®^ 2018 (SPMS EPE, Lisboa, Portugal) and EMR SClínico^®^ 2018 (SPMS EPE, Lisboa, Portugal); and patient telephone surveys using the QueXS^®^ 1.16.0 and 2.2.0 Open Source web-based Computer Assisted Telephone Interview (CATI) system (Australian Consortium for Social and Political Research Incorporated (ACSPRI), Black Rock, Victoria, Australia).

We collected the following: (1) patient case-mix variables (including income status, medication profile, and comorbidities); (2) primary outcome measures; (3) process measures; (4) pharmacy characteristics.

We defined income status using the following: (1) monthly equivalent income per individual calculated from the net monthly household income divided by the number of equivalent adults using the Organization for Economic Co-operation and Development (OECD)-modified age-based scale [49]; (2) patients below at-risk poverty monthly threshold < EUR 501.20 as defined by the National Institute for Statistics (Instituto Nacional de Estatística, or INE) in the database PORDATA 2018.

The number of regular medicines was determined based on the number of different International Non-proprietary Names (INN) used per patient.

The medication profile of antihypertensive and/or lipid-lowering medications first used the World Health Organization (WHO) Anatomical Therapeutic Chemical (ATC) codes and grouped them in medication classes from the International Consortium for Health Outcomes Measurement (ICHOM) Standard Set for Hypertension [50].

For comorbidities, the Rx-Risk Comorbidity Index was used, which lists 43 comorbidity categories, each mapped to a set of prescribed medicines according to the World Health Organization (WHO) Anatomical Therapeutic Chemical (ATC) codes, providing an indirect method to describe patients’ comorbidities based on prescribed medicines in the absence of medical diagnosis data [51].

We also collected cost data, patient-reported outcome measures (PROMs), medication adherence, and patient-reported experience measures (PREMs) for other research studies [30,31].

See Table 1 for a summary of all trial patient measures, time points, and data sources.

We performed a pre-test of telephone surveys. Baseline telephone surveys were administered 24–48 h following recruitment and all telephone surveys were administered according to CATI^®^. We provided training to interviewers to ensure standardization and requested immediate feedback on the first interviews to enable quick adjustments at the onset of the trial. Interviewers were instructed to perform up to 8 tentative calls per patient.

#### 2.5.3. Data Management

We performed data linkage of all real-world datasets through a unique patient identifier (NHS number) and using a common data model that organizes data from different sources that are in different formats into a standard structure. We applied data quality control procedures for data cleaning. We used an encrypted and anonymized file for the analysis.

Data management was performed in compliance with the protocol, standards, and legislation, including the GDPR.

### 2.6. Sample Size

#### 2.6.1. Sample Size Calculation

Sample size calculation was based on the difference in change in BP and TC from baseline to 6 months between the intervention and control, considering a 20% drop-out rate (loss to follow-up), 80% power, and a 5% significance level (two-tailed) to detect change.

We used the literature to guide assumptions for sample size estimates [13,14,16,19,23,52,53,54,55,56].

Assuming a 7 mm Hg difference in BP decrease and a 20 mm Hg standard deviation (SD), 322 hypertensive patients (161 per arm) were needed.

Assuming a 10.5 mg/dL difference in TC decrease and a 32 mg/dL SD, 366 hyperlipidemia patients (183 per arm) were needed.

When looking at clinical significance, a difference of only 5 mmHg in systolic blood pressure (SBP) decrease is clinically meaningful [57]. Every 1 mmol/L (38.7 mg/dL) decrease in TC is clinically meaningful [58].

We planned intervention pharmacies to recruit approx. 48 patients per intervention pharmacy considering a conservative expected proportion of 0.55 of hypertension patients with hyperlipidemia. We planned control pharmacies to recruit approx. 18 patients per control pharmacy given the higher number of control vs. intervention pharmacies.

Based on the number of pharmacies, average number of pharmacists per pharmacy, average sales of hypertension/hyperlipidemia medication per pharmacy per month, and proportion of uncontrolled hypertension and hyperlipidemia patients in Portugal, and on the assumption that 70% of eligible patients usually consent to participate in a study, 6 months was the estimated time frame to recruit the target number of patients (~8/3 patients per month in intervention/control pharmacies).

#### 2.6.2. Interim Analyses

A Data Monitoring Committee was not required due to the short-term nature of the trial and that the intervention was not expected to cause harm. Early trial termination was also not foreseen [59]. We planned interim analyses to allow for earlier adjustments [60] in ICPs, a time window for follow-up assessments at the pharmacy, and statistical methods. Interim analyses (process indicators) started 3 months after the last patient enrollment date.

#### 2.6.3. Withdrawal Procedures

Patients could withdraw consent to participate in the study at any time without compromising their routine care and were not replaced. This was specified in the Study Patient Consent Form. Researchers could also withdraw recruited non-eligible patients.

We considered withdrawal in sample size calculations. All data collected until the loss to follow-up were used for analysis.

### 2.7. Other Indicators

We also assessed the following: (1) process indicators; (2) adverse events; (3) pharmacy profiling.

Process indicators included the following: the number of pharmacy, medical, and nurse appointments, the number of patients referred for a medical appointment, and the number of days between referral and medical appointment.

Adverse events of antihypertensive and/or lipid lowering medicines were assessed.

The profiles of intervention and control pharmacies were compared based on potential differential pharmacy operations and behavioral characteristics capable of influencing the engagement rate in collaborative care: ratio of pharmacists/pharmacy, daily dispensed medicines per pharmacist, number of operating hours per week, engagement of pharmacy director in local or regional leadership, active status in Kaizen™ program, engagement in at least one national or regional professional patient care initiative, active status in Customer Loyalty Program, and active status in refill text reminder program (MED180^®^) which operates in the pharmacy software Sifarma^®^ 2000 2.9.4.3 (Glintt, Sintra, Portugal). Kaizen™ is a Japanese worldwide recognized concept and methodology for continuous improvement of a business by redesigning workflows and improving standardized processes aiming at eliminating waste and redundancies. The Kaizen™ program for pharmacies is available for interested pharmacies in Portugal. Pharmacies operating under Kaizen™ are usually engaged in innovative roles and initiatives.

### 2.8. Assignment Method

We planned for intervention and control pharmacies to recruit patients with an allocation ratio of 1:1.

We used matched control municipalities and PCUs to minimize the potential bias of non-randomization.

### 2.9. Blinding

Due to the behavioral nature of the intervention, it was not possible to blind intervention PCU, pharmacies, or patients. Controls were blinded by design.

### 2.10. Statistical Methods

#### 2.10.1. Descriptive Statistics

We used descriptive statistics to describe patient socio-demographics, baseline clinical and treatment variables, process indicators, and pharmacy profiling.

Continuous data were summarized using mean and SD and median and inter-quartile range (IQR). Categorical data were presented using frequencies and percentages.

#### 2.10.2. Primary Outcomes

We used the outcomes from the primary care EMR database measured by PCU GPs and nurses for the statistical analysis.

We first performed a bivariate between-group analysis using the Wilcoxon–Mann–Whitney and the Chi-square or Fisher Exact Tests, the Wilcoxon Signed-Rank Test for within-group differences, and McNemar’s Test for differences in proportions.

We then used matched controls with two methods to assess changes in outcomes.

We assessed changes from baseline to 6 months using difference-in-differences (DiD) estimators in generalized linear models (GLMs) at these two time points. Model assumptions were checked, and diagnostics were reported. Adjusted models included other covariates imbalanced between the groups at baseline, assumed time-invariant.

Statistical significance was assessed at the 5% (two-sided) level.

We also explored the use of controlled interrupted time series (CITS) [61,62] for BP 6 months before and 6 months following the onset of recruitment. We checked the model and performed sensitivity analysis over a broader time horizon.

The following equation describes the general specification of the DiD model:Y_it_ = β_0_ + β_time_ × Time + β_intervention_ × Intervention + β_interaction_ × Time × Intervention + ε_it_(1)
where the components are as follows:

Y is the outcome of interest (systolic BP, diastolic BP, and TC).

Time (if dichotomic) is 0 = baseline; 1 = 6 months.

Intervention (dichotomic) is 0 = control group (G2); 1 = intervention group (G1).

In addition, β_0_ is the intercept, the mean Outcome A in the G2 group at baseline; β_time_ is the change in Outcome A in the G2 group between baseline and 6 months; the coefficient β_intervention_ is the difference in Outcome A between G1 and G2 groups at baseline; and, finally, β_interaction_ is the difference in slopes between the two groups. In other words, it is the measure of the difference-in-differences between the two groups used to assess the impact of the intervention.

#### 2.10.3. Subgroup Analyses

We performed subgroup GLM analyses of uncontrolled patients at baseline and of low-income deprived patients to assess whether the effect of the intervention varied by each subgroup.

#### 2.10.4. Missing Data

Handling of missing data in the primary analysis consisted of available-case analysis, assuming missing at random (MAR), and we tested the MAR assumption.

Statistical analyses for matching and primary outcome assessment were conducted using R version 3.5.1. (The R Foundation for Statistical Computing, Vienna, Austria) and SAS^®^ Enterprise Guide^®^ 4.2 (SAS Institute Inc, Cary, NC, USA). Statistical analysis of patient-reported variables used IBM^®^ SPSS^®^ Statistics 27 (IBM, New York, NY, USA).

## 3. Results

### 3.1. Primary Care Units and Pharmacies

A total of seven Quality Circles involving 27 pharmacists from seven intervention pharmacies, 6 GPs, and 6 nurses from intervention PCUs took place between June 2016 and May 2019.

We selected 20 potential control PCUs and narrowed them down to the 5 best-match control PCUs. A total of 13 control pharmacies (28%) agreed to participate.

Patient recruitment started at the end of May 2018.

### 3.2. Participant Flow

#### 3.2.1. Patient Flow

A total of 302 patients were invited: 214 accepted and 203 (131 intervention, 72 control) patients entered the study and were included in the baseline analysis for self-reported data in the telephone survey. The number of patients in the primary care database was 107 (73 intervention, 34 control) and 114 (80 intervention, 34 control) for the 6 months before and after recruitment and the start of the intervention, respectively.

We could not find outcome data for the remaining 89 patients in the primary care database. This may have happened because the last medical appointment of these patients occurred 12 months before the onset of the trial, because no blood pressure or total cholesterol was measured for these patients at the PCU in that period, or due to some database error from daily operations and/or data refresh run by the SPMS (this could not be verified by the research team).

Figure 3 provides an overview of patient flow throughout each stage of the study.

#### 3.2.2. Protocol Deviations

After we successfully implemented IT developments for data exchange between the pharmacy dispensing claims software Sifarma^®^ 2000 2.9.4.3 (Glintt, Sintra, Portugal) and the EMR software Vitacare^®^ 2016 (HIS, Lisboa, Portugal) used by the intervention PCU at that time, this was live tested by GPs, nurses, and pharmacists during the fifth Quality Circle in September 2016, and the onset of recruitment was set for November 2016. However, the EMR software private proprietor HIS unexpectedly plead insolvency in Portugal in 2017. The SPMS then replaced Vitacare^®^ 2016 (HIS, Lisboa, Portugal) with their public EMR software SClínico^®^ (SPMS EPE, Lisboa, Portugal) in the intervention PCU in June 2017 and all data exchange interface processes had to be redesigned in the new EMR software. The recruitment started 18 months later.

The collaborative intervention IT components—exchange of point-of-care measurements, request for a repeat prescription, request for a medical appointment, feedback from GP—were, however, either available extremely late or with limitations in the new primary care EMR software SClínico^®^ 2018 (SPMS EPE, Lisboa, Portugal) for most of the trial.

Uncontrolled patients at baseline became no longer a patient compulsory inclusion criterion due to difficulties experienced by pharmacists in identifying baseline uncontrolled patients at recruitment after the 18-month delay.

Five Quality Circles took place in 2016, none in 2017, and two at the onset of recruitment.

All seven Quality Circles took place on weekdays in the evening with a self-service standing welcome dinner in an informal environment as an icebreaker between primary care GPs, nurses, and pharmacists. The work was conducted for 3 hours by the PCU physician coordinator, trial monitor, and lead researcher, and co-chaired by the project physician leader and a board member of the National Association of Pharmacies.

The first Quality Circle started with a motivation/team-building speaker followed by a presentation of teams, a brief description of the project, a discussion, and a wrap-up.

Five more Quality Circles took place in the planned onset year (2016), addressing adjustments in ICPs, live testing data exchange with simulated patient case studies, and presenting the Feedback Reports to enhance recruitment and retention and improve fidelity.

Following the insolvency of the primary care software proprietor HIS, no Quality Circles took place. The last two Quality Circles took place at the onset of the trial to present the major features of the data exchange in the new primary care EMR software. As most IT features were now limited, this affected providers’ engagement.

### 3.3. Recruitment

Patient recruitment took place from May to 30 November 2018 and follow-up data collection ended on 31 July 2019 (6 ± 2 months after the last recruited patient).

### 3.4. Baseline Data and Baseline Equivalence

Intervention patients are more economically deprived, without formal education, and older than the national average. The proportion of patients with ICPC-2 codes K86 and K87 diagnosis (hypertension), T93 diagnosis (lipid metabolism disorder), and T82 diagnosis (obesity) is similar to the national average.

Intervention and control patients were similar at baseline for most characteristics, as expected: gender, employment status, smoking status, most comorbidities, number of regular medicines per patient, number of years since onset, number and class of antihypertensive agent, and class of lipid-lowering medicine (*p* ≥ 0.05). Intervention patients have more education years, higher income, lower Body Mass Index (BMI), and are less likely to be on simultaneous antihypertensive and lipid-lowering medications (*p* < 0.05). These variables were covariates in the multivariate analysis (see Table 2).

### 3.5. Numbers Analyzed

We report the number of participants and considered an intention-to-treat (ITT) population.

### 3.6. Outcomes and Estimation

#### 3.6.1. Using Difference-in-Differences in GLM

The bivariate analysis showed that intervention and control patients were similar at baseline regarding BP and TC, and most patients were already controlled at baseline. We observed no change from baseline in outcomes (*p* > 0.05).

After adjusting the effect of intervention vs. control group for covariates there was still no meaningful change in systolic BP (7.48 mm Hg; 95% CI, −4.17 to 19.33) or TC (22.12 mg/dL; 95% CI, −15.37 to 59.61). The wide CIs express the uncertainty.

The adjusted odds ratio for intervention patients achieving BP control at 6 months is not statistically significant.

We could not observe significant differences in the subgroup of uncontrolled patients at baseline or in the subgroup of most income-deprived patients either (Table 3).

We found no significant difference between patients with missing and completed 6-month BP and/or TC levels after assessing for MAR assumption.

#### 3.6.2. Using Controlled Interrupted Time Series (CITS)

When using a CITS, the trend effect of the intervention vs. control group in systolic BP change, although negative (−0.43 mm Hg), is not significant (95% CI, −4.93 to 4.07) (Figure 4).

We can also observe a downward trend effect (improvement) that seems more pronounced in the 6 months preceding the onset of recruitment for both groups.

The same observation applies to diastolic BP change (0.48 mm Hg; 95% CI, −2.00 to 2.96) (Figure 5).

In the sensitivity analysis using results from the 12 months before and after intervention, the trend effect remains not significant for systolic BP (0.40 mm Hg; 95% CI, −2.22 to 1.41) or diastolic BP (0.29 mm Hg; 95% CI, −0.65 to 1.23), although the CI is narrower.

### 3.7. Process Indicators

The mean (SD) number of pharmacy appointments per intervention patient was just 1.9 (1.1), below what was expected under the ICP. Intervention pharmacies requested a medical appointment for 25 (19%) patients, of which 56% were within 30 days. Based on PCU-reported data for each patient, the mean (SD) number of days between the referral and medical appointment per patient was 57.7 (50.4). For patients with a medical appointment requested within 30 days, the mean number (SD) was 73.1 (57.4), exceeding by 2.4 times the pre-agreed time interval in the ICP.

These process indicators do not exist for control patients as no intervention was in place.

### 3.8. Adverse Events

Adverse events or unintended effects from the intervention itself were not reported.

Pharmacists reported adverse drug events, and the research team reported them to the National Pharmacovigilance System of the Portuguese Medicines Agency INFARMED in 2018 for three patients: one intervention patient, a female aged 61 (atorvastatin 20 mg); two control patients, one male aged 68 (furosemide 40 mg, trandolapril 2 mg, and amlodipine 10 mg) and one female (amlodipine + valsartan).

### 3.9. Pharmacy Profiling

Intervention and control pharmacies were similar in all potential differential operation and behavioral characteristics, except for the mean daily volume of dispensed medicines per pharmacist and the active status in the refill text reminder program (Table 4).

## 4. Discussion

### 4.1. Summary of Findings

This collaborative trial was not able to establish effectiveness versus usual care, unlike other effectiveness trials in hypertension management conducted in community pharmacies [19,23]. We discuss further this study’s limitations that may explain the possible reasons for not achieving our goal, as well as its strengths and future implications for research, policy, and practice.

#### 4.1.1. Limitations

We could not achieve a large enough sample size to rule out uncertainty. The low sample size was due to difficulty in recruitment, despite extending the recruitment to one more month (7 months in total). This may have been caused by the unexpected change in the primary care software, as further explained.

The unexpected change in the primary care software caused an 18-month delay in the onset of recruitment, due to an adjustment period, data migration, permissions from the new software provider, and a redesign of all interface processes required for the IT-driven collaborative care workflow between intervention pharmacies and the PCU. The delay in the planned onset of recruitment had further consequences.

We started workshops and Quality Circles in 2016 (expected onset of recruitment). However, the actual onset of recruitment occurred in 2018. During this gap period, the government enacted national legislation creating specialized family health nurses in PCUs and performance incentives for nurses [63,64,65]. These incentives may have prompted immediate practice improvements by nurses in 2017 both in intervention and control groups. In fact, despite non-significance, we observed a downward trend in blood pressure (improvement) that was more pronounced in the 6 months preceding the onset of recruitment (end of 2017/beginning of 2018), which may help to explain why the proportion of controlled patients was already high at baseline recruitment. In addition, we must consider the possible Hawthorne effect in the intervention group with the first Quality Circles in 2016, which may have prompted the downward trend in blood pressure in the 6 months preceding recruitment, thus also explaining the higher proportion of controlled patients at baseline in 2018. In other words, intervention pharmacists may have behaved differently (e.g., improving care) between 2016 and 2018 because they knew they were being observed.

As a result of the challenges encountered in the real-world trial, we could not achieve a large enough sample size as planned per protocol. These challenges may have also lowered the potential for further improvement as a higher proportion of uncontrolled patients eligible for recruitment may have existed earlier in 2016 but no longer in 2018, again visible in the downward trend in blood pressure before the onset of the recruitment.

The interoperability issues with the new EMR software SClínico^®^ affected the decision algorithms containing the Integrated Care Pathways, which may have also led to a less powerful intervention, although this is not visible with a low sample size.

It was not possible to randomly select the intervention PCU, and this may have caused selection bias issues which we sought to address by using matched controls, control of covariates, and CITS. Future similar non-randomized real-world trials could also benefit from going one step further and exploring the use of a CITS with synthetic controls, an optimization procedure using the outcome variables from the potential controls and any predictor variables to select the best weighting of control units such that the level and trend of the preintervention synthetic control most closely match the intervention [66].

We realize other technical patient inclusion criteria, such as smartphones and online registration to consent to data exchange, may have caused selection bias, but this was part of the intervention package.

Overall, looking at the mechanism by which the intervention was intended to work, we identify weaknesses in three of Wagner’s Chronic Care Model components: clinical information systems, delivery system design, and community resources. Furthermore, the limitations experienced in this trial negatively affected providers’ and patients´ engagement, behavior change, and improvements in outcomes of Blalock’s IMB framework.

#### 4.1.2. Strengths

From a research perspective, this study had some strengths, namely the use of a pragmatic controlled design to assess effectiveness, real-world evidence from various data sources [34,35,36,37,60], intent-to-treat analyses, control of baseline covariates [32], matched controls with a CITS [61,62], a comorbidity algorithm derived from prescribed medicines to classify comorbidities, which is useful in the absence of diagnosis data [51], an impact assessment on equity, and economic and qualitative data collected alongside this trial [29]. We did not match at the pharmacy level, but we checked for differences between intervention and control pharmacies using operating and behavioral characteristics.

From a policy and practice perspective, this first collaborative trial presents a coherent set of tentative innovative strategies on all five levels, designed to foster real-world integrated care [67]: the first collaborative technology-driven trial between pharmacies and primary care providers (administrative); a Privacy Impact Assessment as per the GDPR (administrative); consensus-based care pathways integrated into pharmacy software and, hence, into the pharmacy daily workflow (clinical); options for patients to consent to data sharing involving community pharmacists are now available in the Consent Form of the NHS Patient EHR (organizational); the kick-off of work in technology communication between pharmacies and NHS primary care providers (service delivery); refill SMS reminders to patients (clinical); Quality Circles (organizational); and an experimental risk–share bundled payment model for pharmacies in the trial (funding) to mimic the desired real-world practice.

Future pharmacy collaborative real-world trials should aim at PCUs with capitation or bundled-payment models or, in the case of Portugal, models that use a pay-for-performance component (PCUs model “B”) since these PCUs have financial incentives to reach quality targets and may provide more realistic opportunities to expand collaborative care with pharmacies [68].

Finally, the Decree-Law no. 62/2016, a national binding legislation enacted in 2016, contains the first reference to the possibility of contractual arrangements with Portuguese pharmacies to provide a defined scope of collaborative services with primary care providers and remunerate them if they are effective and cost effective, representing an opportunity to expand pharmacy collaborative real-world trials with primary care providers using the lessons learned from this trial [69].

## 5. Conclusions

Considering the many limitations described, our findings are not generalizable for community pharmacies and primary care providers in Portugal.

This first collaborative trial was not able to show effectiveness versus usual care, most likely due to unexpected changes in the primary care EMS software which affected the sample size and interoperability.

Nevertheless, this project offers valuable lessons on tentative exploratory innovative methods, strategies, and real-world evidence from various data sources, paving the way for advancing integrated care between pharmacies and primary care providers through the use of more real-world effectiveness trials capable of using digital health, interoperability between systems, combined health targets, and performance incentives to close the gap between efficacy and effectiveness.

This is required to improve patient outcomes at affordable costs to the health system towards the goal of value-based healthcare.

## Figures and Tables

**Figure 1 ijerph-20-06496-f001:**
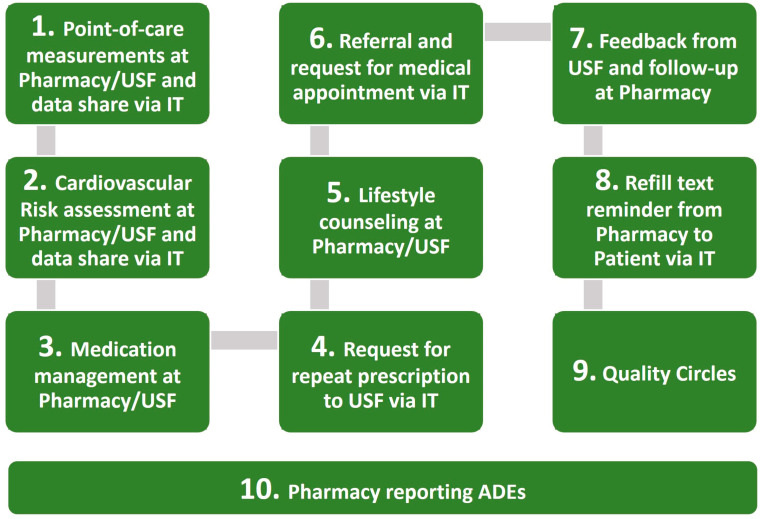
Collaborative care intervention package between intervention PCU and pharmacies.

**Figure 2 ijerph-20-06496-f002:**
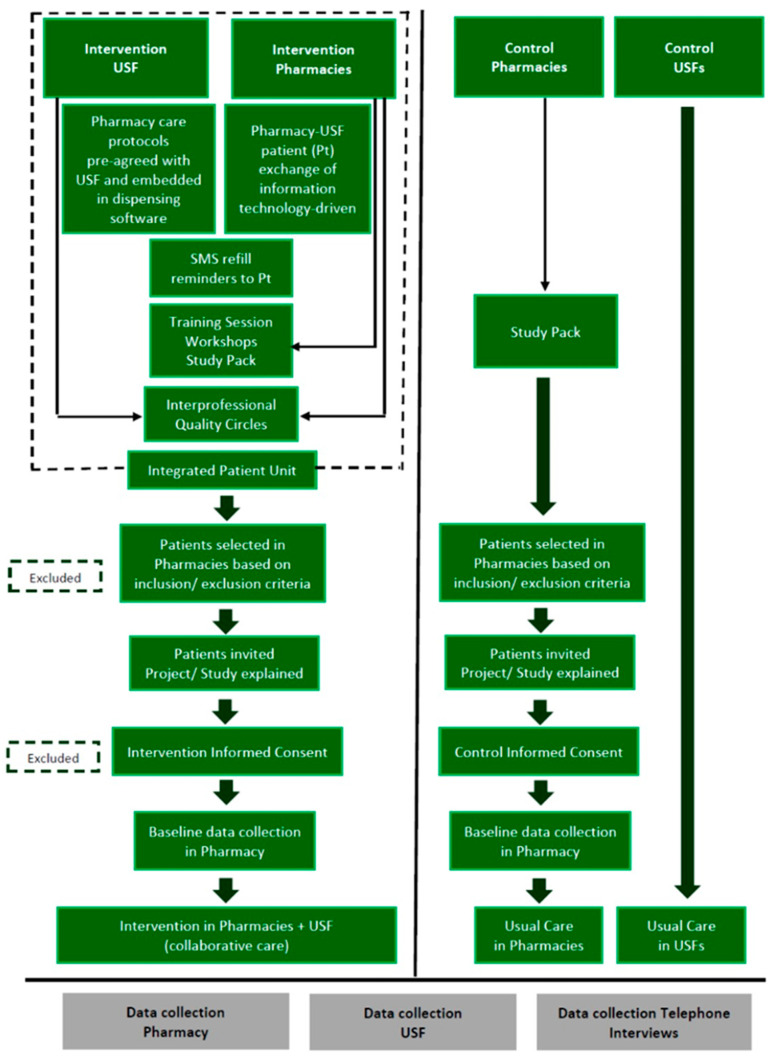
Study flow diagram. Intervention and control PCUs (‘Unidades de Saúde Familiar’, USFs) and pharmacies.

**Figure 3 ijerph-20-06496-f003:**
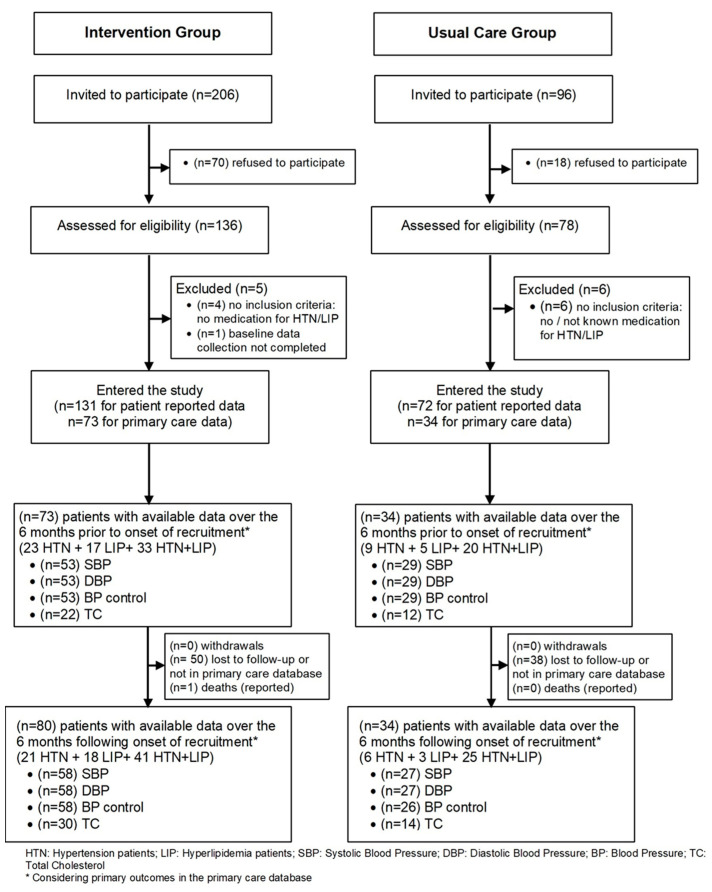
Flowchart of patients throughout each stage of the study.

**Figure 4 ijerph-20-06496-f004:**
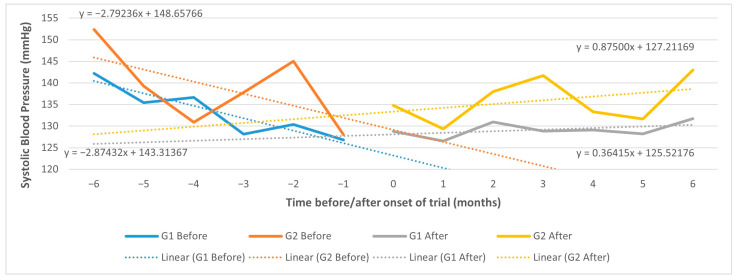
CITS trend effect in systolic BP 6 months before/after; G1: intervention; G2: control.

**Figure 5 ijerph-20-06496-f005:**
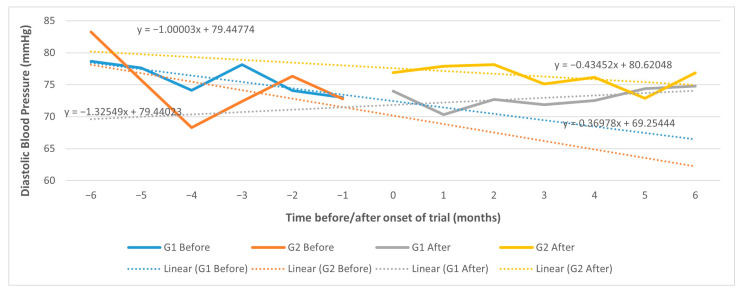
CITS trend effect in diastolic BP 6 months before/after; G1: intervention; G2: control.

**Table 1 ijerph-20-06496-t001:** Summary of all trial measures.

Measures	Timepoint Recorded	Data Sources
**Effectiveness Outcomes**		
*Primary Care* * PCU point-of-care measurements	All available data points: 6 ± 2 M before patient enrollment and 6 ± 2 M after patient enrollment	Primary care Electronic Medical Record (EMR) software (SClínico^®^)
**Process**		
*Pharmacy* * Pharmacy visits Pharmacy point-of-care measurements and tests	All available data points: 6 ± 2 M after patient enrollment (no intervention before enrollment)	Pharmacy dispensing claims software (Sifarma^®^)
*Primary Care* * GP visits Nurse visits PCU point-of-care measurements and tests	All available data points: 6 ± 2 M before patient enrollment and 6 ± 2 M after patient enrollment	Primary care Electronic Medical Record (EMR) software (SClínico^®^)
*Medication* * Prescribed antihypertensive/lipid-lowering medication	All available data points: 6 ± 2 M before patient enrollment and 6 ± 2 M after patient enrollment	Primary care prescription claims software (PEM^®^)
**Economic, Quality, and Preference Outcomes**		
*Quality of Life* ** Quality of life	0 and 6 months	Patient telephone survey (EQ-5D-3L^™^)
*Use of Healthcare Resources* ** Primary care + hospital ER visits Hospital outpatient visits Days in hospital Working days lost Travel + waiting time to PCU/Pharmacy Means of transport + km or cost	0 and 6 months	Patient telephone survey
*Behaviors* ** Medication adherence	All available data points: 6 ± 2 M before patient enrollment and 6 ± 2 M after patient enrollment 0 and 6 months	Primary care prescription claims software (PEM^®^) Patient telephone survey (MAT-7^®^)
*Patient Experience* ** Patient focus group	After the end of the trial	Focus group
*Patient Experience* ** Patient preferences	After the end of the trial	Patient telephone preference survey
*Patient Experience* ** Satisfaction with pharmacy care	0 and 6 months	Patient telephone survey

PCU: Primary Care Unit; GP: General Practitioner; ER: emergency room; M: months. * Collected for this study. ** Collected for other studies.

**Table 2 ijerph-20-06496-t002:** Patient demographics and clinical variables at baseline.

Demographics and Case Mix at Baseline	Intervention (*n* = 131)	Control (*n* = 72)	*p*-Value for Difference (a)
*Gender* (*Telephone baseline survey*)			
Female, *n* (%)	82 (62.6%)	43 (59.7%)	0.6872
Male, *n* (%)	49 (37.4%)	29 (40.3%)	
Age, years (mean ± SD) (*Telephone baseline survey*)	66.4 ± 10.8	64.0 ± 9.8	0.0497
*Education* (*Telephone baseline survey*)			
No. years compulsory education, mean (SD)	9.1 (4.6)	7.5 (4.6)	0.0214
Education ≤ elementary school 3rd cycle (current 9th grade/former 5th grade/technical schools), *n* (%)	64 (58.7%)	45 (75.0%)	0.0343
NR	22	12	
*Employment status* (*Telephone baseline survey*)			
Retired/pensioner + permanently disabled + unemployed + household tasks, *n* (%)	81 (74.3%)	41 (68.4%)	0.4065
NR	22	12	
*Income status (Telephone baseline survey)*			
Approx. monthly equivalent income per person (=household income average threshold/ no. of equalized individuals in household) in EUR (SD) NR = 74	846.30 (569.14)	614.52 (411.54)	0.0058
Approx. monthly household income (= household income average threshold) in EUR (SD) NR = 72	1282.76 (854.84)	943.48 (602.83)	0.0204
≤EUR 501.20 (*n*, %)	23 (26.7%)	22 (48.9%)	0.0113
NR	45	27	
Municipality Purchasing Power Index	95	92,5	
*Smoking status (n, %) (Telephone baseline survey)*			
Smoker (Yes)	11 (9.9%)	5 (8.3%)	0.7355
NR	20	12	
Body Mass Index (mean kg/m^2^ ± SD) (*Telephone baseline survey*)	27.0 ± 4.2	28.4 ± 4.5	0.028l5
*Comorbidities (NR = 7)* *			
No. comorbidities per patient (mean ± SD)	1.9 (1.7)	2.3 (1.8)	0.0988
≥1 (*n*,%)	97 (75.8)	62 (91.2)	
Ischemic heart disease: hypertension	27 (21.1)	27 (39.7)	0.0055
Gastroesophageal reflux disease	29 (22.7)	16 (23.5)	0.8900
Anxiety	29 (22.7)	13 (19.1)	0.5655
Depression	27 (21.1)	13 (19.1)	0.7439
Congestive heart failure	18 (14.1)	12 (17.6)	0.5070
*No. regular medicines per patient (Primary care software)*			
Mean, (SD)	4.5 (2.7)	4.8 (2.8)	0.3916
Minimum–maximum	0–13	0–13	
*Medication profile* *(Pharmacy and primary care software*, *telephone baseline survey*)			
Antihypertensive medication (*n*, %)	44 (33.6%)	19 (26.4%)	0.0117
Lipid-lowering medication (*n*, %)	36 (27.5%)	10 (13.9%)	
Antihypertensive and lipid-lowering medication (*n*, %)	51 (38.9%)	43 (59.7%)	
*Number of years since onset (mean ± SD)* (*Telephone baseline survey*)			
Antihypertensive medication	5.4 (5.6)	6.3 (6.7)	0.8698
Lipid-lowering medication	4.4 (4.4)	5.9 (6.8)	0.4041
*Antihypertensive medication (Primary care software)*			
No. antihypertensive medicines per patient (mean ± SD)	1.5 (0.7)	1.7 (0.9)	0.3950
ACEi/ARB (C09)	83 (64.8%)	51 (75.0%)	0.1456
Alpha-blocker (C02CA)	1 (0.8%)	1 (1.5%)	1.0000
Beta-blocker (C07)	23 (18.0%)	16 (23.5%)	0.3533
Loop diuretics (C03CA + C03CA)	12 (9.4%)	9 (13.2%)	0.4056
Thiazides (C03A)	0 (0.0%)	0 (0.0%)	NA
Calcium channel blocker (C08)	11 (8.6%)	14 (20.6%)	0.0166
Other (C02 + C03)-(C02CA + C03A + C03CA + C03CB)	11 (8.6%)	6 (8.8%)	0.9566
*Lipid-lowering medication (Primary care software)*			
Statin (C10AA + C10BA + C10BX)	67 (52.3%)	38 (55.9%)	0.6363
Ezetimibe (C10AX09)	1 (0.8%)	1 (1.5%)	1.0000
Fibrates (C10AB)	2 (1.6%)	2 (2.9%)	0.6107
Other (C10-(C10AA + C10BA + C10BX + C10AB + C10AX09)	0 (0.0%)	0 (0.0%)	NA

ACEI: Angiotensin-Converting-Enzyme Inhibitor; ARB: Angiotensin II Receptor Blockers; NR: non-respondents; NA: not available. * Top 5 comorbidities presented. (a) Wilcoxon–Mann–Whitney/Chi-square.

**Table 3 ijerph-20-06496-t003:** Unadjusted and adjusted effect of intervention vs. control at 6 months.

Outcome	Model 1 Unadjusted Effect (95% CI)	*p*-Value	Model 2 Adjusted Effect (95% CI) *	*p*-Value
*All patients*				
Systolic BP, mm Hg	−2.08 (−12.09; 7.94)	0.6844	7.48 (−4.17; 19.33)	0.2062
Diastolic BP, mm Hg	−0.42 (−7.74; 6.90)	0.9100	1.66 (−6.46; 9.78)	0.6887
BP control **	1.14 (0.31; 4.24)	0.8477	0.85 (0.17; 4.32)	0.8444
TC, mg/dL	12.84 (−19.70; 45.38)	0.4392	22.12 (−15.37; 59.61)	0.2475
TC control **	0.22 (0.01; 3.17)	0.2629	-	-
*Subgroup uncontrolled at baseline*				
Systolic BP, mm Hg	−2.47 (−16.60; 11.65)	0.7315	5.41 (−8.10; 18.92)	0.4325
Diastolic BP, mm Hg	−1.23 (−13.63; 11.17)	0.8458	1.34 (−13.34; 16.03)	0.8579
TC, mg/dL	44.88 (−4.58; 94.34)	0.0753	36.93 (0.57; 73.28)	0.0465
*Subgroup most income deprived (≤501.20 €/month)*				
Systolic BP, mm Hg	10.88 (−5.17; 26.94)	0.1840	10.85 (−5.77; 27.47)	0.2006
Diastolic BP, mm Hg	6.79 (−4.50; 18.07)	0.2384	6.11 (−5.47; 17.69)	0.3009
BP control **	0.79 (0.07; 8.45)	0.8484	1.05 (0.09; 12.64)	0.9705
TC, mg/dL	19.61 (−26.87; 66.10)	0.4082	−1.41 (−52.64; 49.83)	0.9571
TC control **	-	-	-	-

BP: blood pressure; TC: total cholesterol. Scaled Deviance between 1.02 and 1.62. For BP control and TC control outcome models, Hosmer and Leme show goodness of fit *p* > 0.05 and Wald statistic *p* > 0.05. * Adjustment for baseline covariates (age, level of education, BMI, individual monthly income, most income deprived ≤ EUR 501.20/month). OBS: Subgroup analysis of most income-deprived excluded income covariates. ** Results of the interaction term expressed as the ratio of odds ratio.

**Table 4 ijerph-20-06496-t004:** Profile of intervention and control pharmacies at baseline.

Pharmacy Profiling *	Intervention Pharmacies (*n* = 7)	Control Pharmacies (*n* = 13)	*p*-Value
Ratio of pharmacists/pharmacy, mean (SD)	4.4 (2.6)	3.7 (1.5)	0.6011
Number of daily dispensed POM and NPM per pharmacist, mean (SD)	62.3 (17.6)	122.5 (43.9)	0.0015
Number of operating hours per week, mean (SD)	63.6 (16.2)	76.7 (29.6)	0.1770
Engagement of pharmacy director in local/regional leadership, *n* (%)	1 (14.3)	1 (7.7)	1.0000
Active status in Kaizen™, *n* (%)	5 (71.4)	11 (84.6)	0.5868
Engagement in at least one national/regional professional initiative, *n* (%)	6 (85.7)	12 (92.3)	1.0000
Active status in Pharmacy Customer Loyalty Program Saúda^®^, *n* (%)	5 (71.4)	11 (84.6)	0.5868
Active status in MED180^®^ (refill text reminder), *n* (%)	6 (87.5)	0 (0.0)	0.0001

POM: Prescription-Only Medicines; NPM: Non-Prescription Medicines; * 2018 or last available date provided by ANF, hmR, Farmácias Portuguesas^®^, Portuguese Pharmaceutical Society, and Adjustt, CEFAR.

## Data Availability

The datasets presented in this article are not readily available because the raw data supporting the conclusions of this article contain potentially identifying and sensitive patient information and are deposited in the Institute for Evidence-Based Health (ISBE) data repository. This includes anonymized, non-identifiable patient-level data linked to pharmacy and patient surveys in compliance with privacy regulations. This dataset may be available for unrestricted use to researchers upon request (isbe@isbe.research.ulisboa.pt). Patient-level data linked to primary care was obtained from SPMS. Requests to access this dataset should be directed to https://www.spms.min-saude.pt/.

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
