# Peer review of "Real-World Effectiveness in Hypertension and Hyperlipidemia Collaborative Management between Pharmacies and Primary Care in Portugal: A Multicenter Pragmatic Controlled Trial (USFarmácia®)"

_ijerph, 2023, doi:10.3390/ijerph20156496_

Round 1
Reviewer 1 Report
Dear Authors,
Although the structure of the study is fine, I found the text difficult to follow. It would improve if it were shorter.
In the first paragraph, you mentioned twice that "patients are not controlled". In the next section of the "Introduction," you wrote that patients are treated and followed by GPs. Hence, it looks like they are controlled.
It's okay that you decided on many individual paragraphs. It provides a good layout. Nonetheless, the study turned out challenging to follow for me.
You haven't achieved the goal of the study. Thus the lack of discussion and incomplete conclusions. Please try to work on the trend of the results. The topic is important, and from my point of view, the study and, finally the readers may benefit from presenting a discussion based on the trend.
Reviewer 2 Report
Dear authors,
It was such a pleasure reviewing your excellent work. The topic addressed in this manuscript is extremely relevant to produce evidence of benefits from pharmaceutical intervention, particularly within the scope of primary care.
The topic addressed is original, as it is the first real-world collaborative trial carried out in Portugal with a partnership between pharmacies and primary health care unit, involving several professionals (pharmacists, GPs and nurses).
As mentioned throughout the manuscript, the methodology used in this study is innovative for evaluating the effectiveness of a collaborative intervention, with the intention of producing evidence from a real-world study and not from a traditional explanatory clinical trial.
Despite the detailed description of the proposed methodology, there were some setbacks in the trial implementation process that may have conditioned the expected results of the intervention.
In Chapter 2 (Materials an Methods), line 188 indicates that diabetic patients were excluded. However, the “Patient Information Leaflet (Supplementary File 2)” mentions that patients could be included if they have diabetes (among other circumstances). This should be clarified.
The conclusions presented are consistent with the evidence and arguments presented, and address the main objective for the current study.
The references used are current and appropriate to the context.
Reviewer 3 Report
Thank you for the opportunity to review the above manuscript. This is an interesting study. I have a few comments for the authors to consider:
Introduction:
· Line 97 - Please provide a brief explanation of Wagner’s Chronic Care Model
Material and Method
· Line 133 - How does the Primary care Unit differ to the Primary care Family Health Unit (PCU)?
· Line 166 - What was the rationale for planning to invite 46 control pharmacies and why did the researchers aim to recruit at least 21? How were these numbers determined?
· Line 188 - Why were diabetic patients excluded from the study?
· In Figure 1, it would be helpful to assign the component numbers to each box
· Line 360 - Suggest the support of a statistician for the sample size calculation
Results
· Line 490 – please clarify what is meant by K86 and K87 diagnosis, I could not see an explanation in the methodology.
· Table 1 with the many abbreviations is confusing and should be revised
· Table 3 – what is Kaizen?
Discussion – this requires improvement and clarification of the following:
· Line 590 – why was there a delay in recruitment?
· Line 596 – please explain what is meant by the Hawthorne effect
· Line 647 – what is the Decree-Law?
